# A Wild Bootstrap for Degenerate Kernel Tests

**Kacper Chwialkowski**
Department of Computer Science
University College London
London, Gower Street, WC1E 6BT
`kacper.chwialkowski@gmail.com`

**Dino Sejdinovic**
Gatsby Computational Neuroscience Unit, UCL
17 Queen Square, London WC1N 3AR
`dino.sejdinovic@gmail.com`

**Arthur Gretton**
Gatsby Computational Neuroscience Unit, UCL
17 Queen Square, London WC1N 3AR
`arthur.gretton@gmail.com`

## Abstract

A wild bootstrap method for nonparametric hypothesis tests based on kernel distribution embeddings is proposed. This bootstrap method is used to construct provably consistent tests that apply to random processes, for which the naive permutation-based bootstrap fails. It applies to a large group of kernel tests based on V-statistics, which are degenerate under the null hypothesis, and non-degenerate elsewhere. To illustrate this approach, we construct a two-sample test, an instantaneous independence test and a multiple lag independence test for time series. In experiments, the wild bootstrap gives strong performance on synthetic examples, on audio data, and in performance benchmarking for the Gibbs sampler. The code is available at `https://github.com/kacperChwialkowski/wildBootstrap`.

## 1 Introduction

Statistical tests based on distribution embeddings into reproducing kernel Hilbert spaces have been applied in many contexts, including two sample testing [18, 15, 32], tests of independence [17, 33, 4], tests of conditional independence [14, 33], and tests for higher order (Lancaster) interactions [24]. For these tests, consistency is guaranteed if and only if the observations are independent and identically distributed. Much real-world data fails to satisfy the i.i.d. assumption: audio signals, EEG recordings, text documents, financial time series, and samples obtained when running Markov Chain Monte Carlo, all show significant temporal dependence patterns.

The asymptotic behaviour of kernel test statistics becomes quite different when temporal dependencies exist within the samples. In recent work on independence testing using the Hilbert-Schmidt Independence Criterion (HSIC) [8], the asymptotic distribution of the statistic under the null hypothesis is obtained for a pair of independent time series, which satisfy an absolute regularity or a $\phi$-mixing assumption. In this case, the null distribution is shown to be an infinite weighted sum of *dependent* $\chi^2$-variables, as opposed to the sum of *independent* $\chi^2$-variables obtained in the i.i.d. setting [17]. The difference in the asymptotic null distributions has important implications in practice: under the i.i.d. assumption, an empirical estimate of the null distribution can be obtained by repeatedly permuting the time indices of one of the signals. This breaks the temporal dependence within the permuted signal, which causes the test to return an elevated number of false positives, when used for testing time series. To address this problem, an alternative estimate of the null distribution is proposed in [8], where the null distribution is simulated by repeatedly *shifting* one signal relative to the other. This preserves the temporal structure within each signal, while breaking the cross-signal dependence.

A serious limitation of the shift procedure in [8] is that it is specific to the problem of independence testing: there is no obvious way to generalise it to other testing contexts. For instance, we might have two time series, with the goal of comparing their marginal distributions - this is a generalization of the two-sample setting to which the shift approach does not apply.

We note, however, that many kernel tests have a test statistic with a particular structure: the Maximum Mean Discrepancy (MMD), HSIC, and the Lancaster interaction statistic, each have empirical estimates which can be cast as normalized $V$-statistics, $\frac{1}{n^{m-1}} \sum_{1 \leq i_1, \dots, i_m \leq n} h(Z_{i_1}, \dots, Z_{i_m})$, where $Z_{i_1}, \dots, Z_{i_m}$ are samples from a random process at the time points $\{i_1, \dots, i_m\}$. We show that a method of external randomization known as the *wild bootstrap* may be applied [21, 28] to simulate from the null distribution. In brief, the arguments of the above sum are repeatedly multiplied by random, user-defined time series. For a test of level $\alpha$, the $1 - \alpha$ quantile of the empirical distribution obtained using these perturbed statistics serves as the test threshold. This approach has the important advantage over [8] that it may be applied to *all* kernel-based tests for which $V$-statistics are employed, and not just for independence tests.

The main result of this paper is to show that the wild bootstrap procedure yields consistent tests for time series, i.e., tests based on the wild bootstrap have a Type I error rate (of wrongly rejecting the null hypothesis) approaching the design parameter $\alpha$, and a Type II error (of wrongly accepting the null) approaching zero, as the number of samples increases. We use this result to construct a two-sample test using MMD, and an independence test using HSIC. The latter procedure is applied both to testing for instantaneous independence, and to testing for independence across multiple time lags, for which the earlier shift procedure of [8] cannot be applied.

We begin our presentation in Section 2, with a review of the $\tau$-mixing assumption required of the time series, as well as of $V$-statistics (of which MMD and HSIC are instances). We also introduce the form taken by the wild bootstrap. In Section 3, we establish a general consistency result for the wild bootstrap procedure on $V$-statistics, which we apply to MMD and to HSIC in Section 4. Finally, in Section 5, we present a number of empirical comparisons: in the two sample case, we test for differences in audio signals with the same underlying pitch, and present a performance diagnostic for the output of a Gibbs sampler (the MCMC M.D.); in the independence case, we test for independence of two time series sharing a common variance (a characteristic of econometric models), and compare against the test of [4] in the case where dependence may occur at multiple, potentially unknown lags. Our tests outperform both the naive approach which neglects the dependence structure within the samples, and the approach of [4], when testing across multiple lags. Detailed proofs are found in the appendices of an accompanying technical report [9], which we reference from the present document as needed.

## 2 Background

The main results of the paper are based around two concepts: $\tau$-mixing [10], which describes the dependence within the time series, and $V$-statistics [27], which constitute our test statistics. In this section, we review these topics, and introduce the concept of wild bootstrapped $V$-statistics, which will be the key ingredient in our test construction.

**$\tau$-mixing.** The notion of $\tau$-mixing is used to characterise weak dependence. It is a less restrictive alternative to classical mixing coefficients, and is covered in depth in [10]. Let $\{Z_t, \mathcal{F}_t\}_{t \in \mathbb{N}}$ be a stationary sequence of integrable random variables, defined on a probability space $\Omega$ with a probability measure $P$ and a natural filtration $\mathcal{F}_t$. The process is called $\tau$-dependent if

$$\tau(r) = \sup_{l \in \mathbb{N}} \frac{1}{l} \sup_{r \leq i_1 \leq \dots \leq i_l} \tau(\mathcal{F}_0, (Z_{i_1}, \dots, Z_{i_l})) \overset{r \to \infty}{\longrightarrow} 0, \text{ where}$$

$$\tau(\mathcal{M}, X) = E\left(\sup_{g \in \Lambda} \left| \int g(t) P_{X|\mathcal{M}}(dt) - \int g(t) P_X(dt) \right| \right)$$

and $\Lambda$ is the set of all one-Lipschitz continuous real-valued functions on the domain of $X$. $\tau(\mathcal{M}, X)$ can be interpreted as the minimal $L_1$ distance between $X$ and $X^*$ such that $X \overset{d}{=} X^*$ and $X^*$ is independent of $\mathcal{M} \subset \mathcal{F}$. Furthermore, if $\mathcal{F}$ is rich enough, this $X^*$ can be constructed (see Proposition 4 in the Appendix). More information is provided in the Appendix B.

*V*-**statistics.** The test statistics considered in this paper are always $V$-statistics. Given the observations $Z = \{Z_t\}_{t=1}^n$, a $V$-statistic of a symmetric function $h$ taking $m$ arguments is given by

$$V(h, Z) = \frac{1}{n^m} \sum_{i \in N^m} h(Z_{i_1}, ..., Z_{i_m}), \tag{1}$$

where $N^m$ is a Cartesian power of a set $N = \{1, ..., n\}$. For simplicity, we will often drop the second argument and write simply $V(h)$.

We will refer to the function $h$ as to the *core* of the $V$-statistic $V(h)$. While such functions are usually called kernels in the literature, in this paper we reserve the term kernel for positive-definite functions taking two arguments. A core $h$ is said to be $j$-degenerate if for each $z_1, \ldots, z_j$ $Eh(z_1, \ldots, z_j, Z_{j+1}^*, \ldots, Z_m^*) = 0$, where $Z_{j+1}^*, \ldots, Z_m^*$ are independent copies of $Z_1$. If $h$ is $j$-degenerate for all $j \leq m - 1$, we will say that it is *canonical*. For a one-degenerate core $h$, we define an auxiliary function $h_2$, called the second component of the core, and given by $h_2(z_1, z_2) = Eh(z_1, z_2, Z_3^*, \ldots, Z_m^*)$. Finally we say that $nV(h)$ is a normalized $V$-statistic, and that a $V$-statistic with a one-degenerate core is a degenerate $V$-statistic. This degeneracy is common to many kernel statistics when the null hypothesis holds [15, 17, 24].

Our main results will rely on the fact that $h_2$ governs the asymptotic behaviour of normalized degenerate $V$-statistics. Unfortunately, the limiting distribution of such $V$-statistics is quite complicated - it is an infinite sum of *dependent* $\chi^2$-distributed random variables, with a dependence determined by the temporal dependence structure within the process $\{Z_t\}$ and by the eigenfunctions of a certain integral operator associated with $h_2$ [5, 8]. Therefore, we propose a bootstrapped version of the $V$-statistics which will allow a consistent approximation of this difficult limiting distribution.

**Bootstrapped *V*-statistic.** We will study two versions of the bootstrapped $V$-statistics

$$V_{b1}(h, Z) = \frac{1}{n^m} \sum_{i \in N^m} W_{i_1,n} W_{i_2,n} h(Z_{i_1}, ..., Z_{i_m}), \tag{2}$$

$$V_{b2}(h, Z) = \frac{1}{n^m} \sum_{i \in N^m} \tilde{W}_{i_1,n} \tilde{W}_{i_2,n} h(Z_{i_1}, ..., Z_{i_m}), \tag{3}$$

where $\{W_{t,n}\}_{1 \leq t \leq n}$ is an auxiliary wild bootstrap process and $\tilde{W}_{t,n} = W_{t,n} - \frac{1}{n} \sum_{j=1}^n W_{j,n}$. This auxiliary process, proposed by [28, 21], satisfies the following assumption:

*Bootstrap assumption:* $\{W_{t,n}\}_{1 \leq t \leq n}$ is a row-wise strictly stationary triangular array independent of all $Z_t$ such that $EW_{t,n} = 0$ and $\sup_n E|W_{t,n}^{2+\sigma}| < \infty$ for some $\sigma > 0$. The autocovariance of the process is given by $EW_{s,n}W_{t,n} = \rho(|s - t|/l_n)$ for some function $\rho$, such that $\lim_{u \to 0} \rho(u) = 1$ and $\sum_{r=1}^{n-1} \rho(|r|/l_n) = O(l_n)$. The sequence $\{l_n\}$ is taken such that $l_n = o(n)$ but $\lim_{n \to \infty} l_n = \infty$. The variables $W_{t,n}$ are $\tau$-weakly dependent with coefficients $\tau(r) \leq C\zeta^{\frac{r}{l_n}}$ for $r = 1, ..., n$, $\zeta \in (0, 1)$ and $C \in \mathbb{R}$.

As noted in in [21, Remark 2], a simple realization of a process that satisfies this assumption is $W_{t,n} = e^{-1/l_n} W_{t-1,n} + \sqrt{1 - e^{-2/l_n}} \epsilon_t$ where $W_{0,n}$ and $\epsilon_1, \ldots, \epsilon_n$ are independent standard normal random variables. For simplicity, we will drop the index $n$ and write $W_t$ instead of $W_{t,n}$. A process that fulfils the *bootstrap assumption* will be called *bootstrap process*. Further discussion of the wild bootstrap is provided in the Appendix A. The versions of the bootstrapped $V$-statistics in (2) and (3) were previously studied in [21] for the case of canonical cores of degree $m = 2$. We extend their results to higher degree cores (common within the kernel testing framework), which are not necessarily one-degenerate. When stating a fact that applies to both $V_{b1}$ and $V_{b2}$, we will simply write $V_b$, and the argument $Z$ will be dropped when there is no ambiguity.

## 3 Asymptotics of wild bootstrapped *V*-statistics

In this section, we present main Theorems that describe asymptotic behaviour of $V$-statistics. In the next section, these results will be used to construct kernel-based statistical tests applicable to dependent observations. Tests are constructed so that the $V$-statistic is degenerate under the null hypothesis and non-degenerate under the alternative. Theorem 1 guarantees that the bootstrapped $V$-statistic will converge to the same limiting null distribution as the simple $V$-statistic. Following [21], we will establish the convergence of the bootstrapped distribution to the desired asymptotic

distribution in the Prokhorov metric $\varphi$ [13, Section 11.3]), and ensure that this distance approaches zero *in probability* as $n \to \infty$. This two-part convergence statement is needed due to the additional randomness introduced by the $W_{j,n}$.

**Theorem 1.** *Assume that the stationary process $\{Z_t\}$ is $\tau$-dependent with $\tau(r) = O(r^{-6-\epsilon})$ for some $\epsilon > 0$. If the core $h$ is a Lipschitz continuous, one-degenerate, and bounded function of $m$ arguments and its $h_2$-component is a positive definite kernel, then $\varphi(n\binom{m}{2}V_b(h, Z), nV(h, Z)) \to 0$ in probability as $n \to \infty$, where $\varphi$ is Prokhorov metric.*

*Proof.* By Lemma 3 and Lemma 2 respectively, $\varphi(nV_b(h), nV_b(h_2))$ and $\varphi(nV(h), n\binom{m}{2}V(h_2))$ converge to zero. By [21, Theorem 3.1], $nV_b(h_2)$ and $nV(h_2, Z)$ have the same limiting distribution, i.e., $\varphi(nV_b(h_2), nV(h_2, Z)) \to 0$ in probability under certain assumptions. Thus, it suffices to check these assumptions hold: *Assumption A2.* (i) $h_2$ is one-degenerate and symmetric - this follows from Lemma 1; (ii) $h_2$ is a kernel - is one of the assumptions of this Theorem; (iii) $Eh_2(Z_1, Z_1) \le \infty$ - by Lemma 7, $h_2$ is bounded and therefore has a finite expected value; (iv) $h_2$ is Lipschitz continuous - follows from Lemma 7. *Assumption B1.* $\sum_{r=1}^{n} r^2 \sqrt{\tau(r)} < \infty$. Since $\tau(r) = O(r^{-6-\epsilon})$ then $\sum_{r=1}^{n} r^2 \sqrt{\tau(r)} \le C \sum_{r=1}^{n} r^{-1-\epsilon/2} \le \infty$. *Assumption B2.* This assumption about the auxiliary process $\{W_t\}$ is the same as our *Bootstrap assumption*. $\square$

On the other hand, if the $V$-statistic is not degenerate, which is usually true under the alternative, it converges to some non-zero constant. In this setting, Theorem 2 guarantees that the bootstrapped $V$-statistic will converge to zero in probability. This property is necessary in testing, as it implies that the test thresholds computed using the bootstrapped $V$-statistics will also converge to zero, and so will the corresponding Type II error. The following theorem is due to Lemmas 4 and 5.

**Theorem 2.** *Assume that the process $\{Z_t\}$ is $\tau$-dependent with a coefficient $\tau(r) = O(r^{-6-\epsilon})$. If the core $h$ is a Lipschitz continuous, symmetric and bounded function of $m$ arguments, then $nV_{b2}(h)$ converges in distribution to some non-zero random variable with finite variance, and $V_{b1}(h)$ converges to zero in probability.*

Although both $V_{b2}$ and $V_{b1}$ converge to zero, the rate and the type of convergence are not the same: $nV_{b2}$ converges in law to some random variable while the behaviour of $nV_{b1}$ is unspecified. As a consequence, tests that utilize $V_{b2}$ usually give lower Type II error then the ones that use $V_{b1}$. On the other hand, $V_{b1}$ seems to better approximate $V$-statistic distribution under the null hypothesis. This agrees with our experiments in Section 5 as well as with those in [21, Section 5]).

## 4   Applications to Kernel Tests

In this section, we describe how the wild bootstrap for $V$-statistics can be used to construct kernel tests for independence and the two-sample problem, which are applicable to weakly dependent observations. We start by reviewing the main concepts underpinning the kernel testing framework.

For every symmetric, positive definite function, i.e., *kernel* $k : \mathcal{X} \times \mathcal{X} \to \mathbb{R}$, there is an associated reproducing kernel Hilbert space $\mathcal{H}_k$ [3, p. 19]. The kernel embedding of a probability measure $P$ on $\mathcal{X}$ is an element $\mu_k(P) \in \mathcal{H}_k$, given by $\mu_k(P) = \int k(\cdot, x)\, dP(x)$ [3, 29]. If a measurable kernel $k$ is bounded, the mean embedding $\mu_k(P)$ exists for all probability measures on $\mathcal{X}$, and for many interesting bounded kernels $k$, including the Gaussian, Laplacian and inverse multi-quadratics, the kernel embedding $P \mapsto \mu_k(P)$ is injective. Such kernels are said to be *characteristic* [31]. The RKHS-distance $\|\mu_k(P_x) - \mu_k(P_y)\|^2_{\mathcal{H}_k}$ between embeddings of two probability measures $P_x$ and $P_y$ is termed the Maximum Mean Discrepancy (MMD), and its empirical version serves as a popular statistic for non-parametric two-sample testing [15]. Similarly, given a sample of paired observations $\{(X_i, Y_i)\}_{i=1}^{n} \sim P_{xy}$, and kernels $k$ and $l$ respectively on $X$ and $Y$ domains, the RKHS-distance $\|\mu_\kappa(P_{xy}) - \mu_\kappa(P_x P_y)\|^2_{\mathcal{H}_\kappa}$ between embeddings of the joint distribution and of the product of the marginals, measures dependence between $X$ and $Y$. Here, $\kappa((x, y), (x', y')) = k(x, x')l(y, y')$ is the kernel on the product space of $X$ and $Y$ domains. This quantity is called Hilbert-Schmidt Independence Criterion (HSIC) [16, 17]. When characteristic RKHSs are used, the HSIC is zero iff $X \perp\!\!\!\perp Y$: this follows from [22, Lemma 3.8] and [30, Proposition 2]. The empirical statistic is written $\widehat{\mathrm{HSIC}}_\kappa = \frac{1}{n^2}\mathrm{Tr}(KHLH)$ for kernel matrices $K$ and $L$ and the centering matrix $H = I - \frac{1}{n}\mathbf{1}\mathbf{1}^\top$.

## 4.1 Wild Bootstrap For MMD

Denote the observations by $\{X_i\}_{i=1}^{n_x} \sim P_x$, and $\{Y_j\}_{j=1}^{n_y} \sim P_y$. Our goal is to test the null hypothesis $\mathbf{H}_0 : P_x = P_y$ vs. the alternative $\mathbf{H}_1 : P_x \neq P_y$. In the case where samples have equal sizes, i.e., $n_x = n_y$, application of the wild bootstrap to MMD-based tests on dependent samples is straightforward: the empirical MMD can be written as a $V$-statistic with the core of degree two on pairs $z_i = (x_i, y_i)$ given by $h(z_1, z_2) = k(x_1, x_2) - k(x_1, y_2) - k(x_2, y_1) + k(y_1, y_2)$. It is clear that whenever $k$ is Lipschitz continuous and bounded, so is $h$. Moreover, $h$ is a valid positive definite kernel, since it can be represented as an RKHS inner product $\langle k(\cdot, x_1) - k(\cdot, y_1), k(\cdot, x_2) - k(\cdot, y_2) \rangle_{\mathcal{H}_k}$. Under the null hypothesis, $h$ is also one-degenerate, i.e., $Eh((x_1, y_1), (X_2, Y_2)) = 0$. Therefore, we can use the bootstrapped statistics in (2) and (3) to approximate the null distribution and attain a desired test level.

When $n_x \neq n_y$, however, it is no longer possible to write the empirical MMD as a one-sample $V$-statistic. We will therefore require the following bootstrapped version of MMD

$$\widehat{\text{MMD}}_{k,b} = \frac{1}{n_x^2} \sum_{i=1}^{n_x} \sum_{j=1}^{n_x} \tilde{W}_i^{(x)} \tilde{W}_j^{(x)} k(x_i, x_j) - \frac{1}{n_x^2} \sum_{i=1}^{n_y} \sum_{j=1}^{n_y} \tilde{W}_i^{(y)} \tilde{W}_j^{(y)} k(y_i, y_j)$$

$$- \frac{2}{n_x n_y} \sum_{i=1}^{n_x} \sum_{j=1}^{n_y} \tilde{W}_i^{(x)} \tilde{W}_j^{(y)} k(x_i, y_j), \tag{4}$$

where $\tilde{W}_t^{(x)} = W_t^{(x)} - \frac{1}{n_x} \sum_{i=1}^{n_x} W_i^{(x)}$, $\tilde{W}_t^{(y)} = W_t^{(y)} - \frac{1}{n_y} \sum_{j=1}^{n_y} W_j^{(y)}$; $\{W_t^{(x)}\}$ and $\{W_t^{(y)}\}$ are two auxiliary wild bootstrap processes that are independent of $\{X_t\}$ and $\{Y_t\}$ and also independent of each other, both satisfying the bootstrap assumption in Section 2. The following Proposition shows that the bootstrapped statistic has the same asymptotic null distribution as the empirical MMD. The proof follows that of [21, Theorem 3.1], and is given in the Appendix.

**Proposition 1.** *Let $k$ be bounded and Lipschitz continuous, and let $\{X_t\}$ and $\{Y_t\}$ both be $\tau$-dependent with coefficients $\tau(r) = O(r^{-6-\epsilon})$, but independent of each other. Further, let $n_x = \rho_x n$ and $n_y = \rho_y n$ where $n = n_x + n_y$. Then, under the null hypothesis $P_x = P_y$, $\varphi\left(\rho_x \rho_y n \widehat{MMD}_k, \rho_x \rho_y n \widehat{MMD}_{k,b}\right) \to 0$ in probability as $n \to \infty$, where $\varphi$ is the Prokhorov metric and $\widehat{MMD}_k$ is the MMD between empirical measures.*

## 4.2 Wild Bootstrap For HSIC

Using HSIC in the context of random processes is not new in the machine learning literature. For a 1-approximating functional of an absolutely regular process [6], convergence in probability of the empirical HSIC to its population value was shown in [34]. No asymptotic distributions were obtained, however, nor was a statistical test constructed. The asymptotics of a normalized $V$-statistic were obtained in [8] for absolutely regular and $\phi$-mixing processes [12]. Due to the intractability of the null distribution for the test statistic, the authors propose a procedure to approximate its null distribution using circular shifts of the observations leading to tests of instantaneous independence, i.e., of $X_t \perp\!\!\!\perp Y_t$, $\forall t$. This was shown to be consistent under the null (i.e., leading to the correct Type I error), however consistency of the shift procedure under the alternative is a challenging open question (see [8, Section A.2] for further discussion). In contrast, as shown below in Propositions 2 and 3 (which are direct consequences of the Theorems 1 and 2), the wild bootstrap guarantees test consistency under both hypotheses: null and alternative, which is a major advantage. In addition, the wild bootstrap can be used in constructing a test for the harder problem of determining independence across multiple lags simultaneously, similar to the one in [4].

Following symmetrisation, it is shown in [17, 8] that the empirical HSIC can be written as a degree four $V$-statistic with core given by

$$h(z_1, z_2, z_3, z_4) = \frac{1}{4!} \sum_{\pi \in S_4} k(x_{\pi(1)}, x_{\pi(2)}) [l(y_{\pi(1)}, y_{\pi(2)}) + l(y_{\pi(3)}, y_{\pi(4)}) - 2l(y_{\pi(2)}, y_{\pi(3)})],$$

where we denote by $S_n$ the group of permutations over $n$ elements. Thus, we can directly apply the theory developed for higher-order $V$-statistics in Section 3. We consider two types of tests: instantaneous independence and independence at multiple time lags.

Table 1: Rejection rates for two-sample experiments. **MCMC**: sample size=500; a Gaussian kernel with bandwidth $\sigma = 1.7$ is used; every second Gibbs sample is kept (i.e., after a pass through both dimensions). **Audio**: sample sizes are $(n_x, n_y) = \{(300, 200), (600, 400), (900, 600)\}$; a Gaussian kernel with bandwidth $\sigma = 14$ is used. **Both**: wild bootstrap uses blocksize of $l_n = 20$; averaged over at least 200 trials. The Type II error for all tests was zero

| | experiment \ method | permutation | $\widehat{\text{MMD}}_{k,b}$ | $V_{b1}$ | $V_{b2}$ |
|---|---|---|---|---|---|
| **MCMC** | i.i.d. vs i.i.d. ($\mathbf{H_0}$) | .040 | .025 | .012 | .070 |
| | i.i.d. vs Gibbs ($\mathbf{H_0}$) | .528 | .100 | .052 | .105 |
| | Gibbs vs Gibbs ($\mathbf{H_0}$) | .680 | .110 | .060 | .100 |
| **Audio** | $\mathbf{H_0}$ | {.970,.965,.995} | {.145,.120,.114} | | |
| | $\mathbf{H_1}$ | {1,1,1} | {.600,.898,.995} | | |

**Test of instantaneous independence**    Here, the null hypothesis $\mathbf{H_0}$ is that $X_t$ and $Y_t$ are independent at all times $t$, and the alternative hypothesis $\mathbf{H_1}$ is that they are dependent.

**Proposition 2.** *Under the null hypothesis, if the stationary process $Z_t = (X_t, Y_t)$ is $\tau$-dependent with a coefficient $\tau(r) = O\left(r^{-6-\epsilon}\right)$ for some $\epsilon > 0$, then $\varphi(6nV_b(h), nV(h)) \to 0$ in probability, where $\varphi$ is the Prokhorov metric.*

*Proof.* Since $k$ and $l$ are bounded and Lipschitz continuous, the core $h$ is bounded and Lipschitz continuous. One-degeneracy under the null hypothesis was stated in [17, Theorem 2], and that $h_2$ is a kernel is shown in [17, section A.2, following eq. (11)]. The result follows from Theorem 1. $\quad\square$

The following proposition holds by the Theorem 2, since the core $h$ is Lipschitz continuous, symmetric and bounded.

**Proposition 3.** *If the stationary process $Z_t$ is $\tau$-dependent with a coefficient $\tau(r) = O\left(r^{-6-\epsilon}\right)$ for some $\epsilon > 0$, then under the alternative hypothesis $nV_{b2}(h)$ converges in distribution to some random variable with a finite variance and $V_{b1}$ converges to zero in probability.*

**Lag-HSIC**    Propositions 2 and 3 also allow us to construct a test of time series independence that is similar to one designed by [4]. Here, we will be testing against a broader null hypothesis: $X_t$ and $Y_{t'}$ are independent for $|t - t'| < M$ for an arbitrary large but fixed $M$. In the Appendix, we show how to construct a test when $M \to \infty$, although this requires an additional assumption about the uniform convergence of cumulative distribution functions.

Since the time series $Z_t = (X_t, Y_t)$ is stationary, it suffices to check whether there exists a dependency between $X_t$ and $Y_{t+m}$ for $-M \leq m \leq M$. Since each lag corresponds to an individual hypothesis, we will require a Bonferroni correction to attain a desired test level $\alpha$. We therefore define $q = 1 - \frac{\alpha}{2M+1}$. The shifted time series will be denoted $Z_t^m = (X_t, Y_{t+m})$. Let $S_{m,n} = nV(h, Z^m)$ denote the value of the normalized HSIC statistic calculated on the shifted process $Z_t^m$. Let $F_{b,n}$ denote the empirical cumulative distribution function obtained by the bootstrap procedure using $nV_b(h, Z)$. The test will then reject the null hypothesis if the event $\mathcal{A}_n = \left\{\max_{-M \leq m \leq M} S_{m,n} > F_{b,n}^{-1}(q)\right\}$ occurs. By a simple application of the union bound, it is clear that the asymptotic probability of the Type I error will be $\lim_{n \to \infty} P_{\mathbf{H_0}}(\mathcal{A}_n) \leq \alpha$. On the other hand, if the alternative holds, there exists some $m$ with $|m| \leq M$ for which $V(h, Z^m) = n^{-1} S_{m,n}$ converges to a non-zero constant. In this case

$$P_{\mathbf{H_1}}(\mathcal{A}_n) \geq P_{\mathbf{H_1}}(S_{m,n} > F_{b,n}^{-1}(q)) = P_{\mathbf{H_1}}(n^{-1}S_{m,n} > n^{-1}F_{b,n}^{-1}(q)) \to 1 \qquad (5)$$

as long as $n^{-1}F_{b,n}^{-1}(q) \to 0$, which follows from the convergence of $V_b$ to zero in probability shown in Proposition 3. Therefore, the Type II error of the multiple lag test is guaranteed to converge to zero as the sample size increases. Our experiments in the next Section demonstrate that while this procedure is defined over a finite range of lags, it results in tests more powerful than the procedure for an infinite number of lags proposed in [4]. We note that a procedure that works for an infinite number of lags, although possible to construct, does not add much practical value under the present assumptions. Indeed, since the $\tau$-mixing assumption applies to the joint sequence $Z_t = (X_t, Y_t)$,

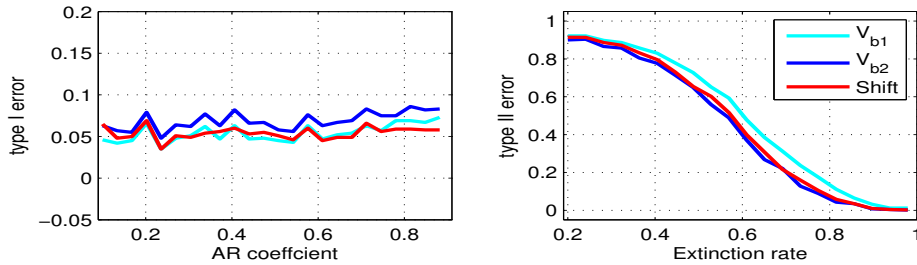

Figure 1: Comparison of Shift-HSIC and tests based on $V_{b1}$ and $V_{b2}$. The left panel shows the performance under the null hypothesis, where a larger AR coefficient implies a stronger temporal dependence. The right panel show the performance under the alternative hypothesis, where a larger extinction rate implies a greater dependence between processes.

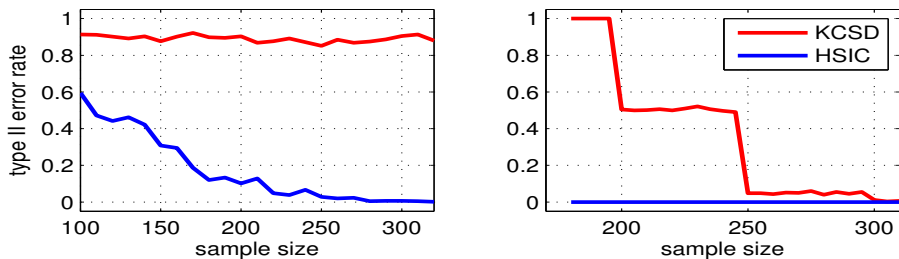

Figure 2: In both panel Type II error is plotted. The left panel presents the error of the lag-HSIC and KCSD algorithms for a process following dynamics given by the equation (6). The errors for a process with dynamics given by equations (7) and (8) are shown in the right panel. The X axis is indexed by the time series length, i.e., sample size. The Type I error was around 5%.

dependence between $X_t$ and $Y_{t+m}$ is bound to disappear at a rate of $o(m^{-6})$, i.e., the variables both within and across the two series are assumed to become gradually independent at large lags.

## 5  Experiments

**The MCMC M.D.**   We employ MMD in order to diagnose how far an MCMC chain is from its stationary distribution [26, Section 5], by comparing the MCMC sample to a benchmark sample. A hypothesis test of whether the sampler has converged based on the standard permutation-based bootstrap leads to too many rejections of the null hypothesis, due to dependence within the chain. Thus, one would require heavily thinned chains, which is wasteful of samples and computationally burdensome. Our experiments indicate that the wild bootstrap approach allows consistent tests directly on the chains, as it attains a desired number of false positives.

To assess performance of the wild bootstrap in determining MCMC convergence, we consider the situation where samples $\{X_i\}$ and $\{Y_i\}$ are bivariate, and both have the identical marginal distribution given by an elongated normal $P = \mathcal{N}\left(\begin{bmatrix} 0 & 0 \end{bmatrix}, \begin{bmatrix} 15.5 & 14.5 \\ 14.5 & 15.5 \end{bmatrix}\right)$. However, they could have arisen either as independent samples, or as outputs of the Gibbs sampler with stationary distribution $P$. Table 1 shows the *rejection rates* under the significance level $\alpha = 0.05$. It is clear that in the case where at least one of the samples is a Gibbs chain, the permutation-based test has a Type I error much larger than $\alpha$. The wild bootstrap using $V_{b1}$ (without artificial degeneration) yields the correct Type I error control in these cases. Consistent with findings in [21, Section 5], $V_{b1}$ mimics the null distribution better than $V_{b2}$. The bootstrapped statistic $\widehat{\mathrm{MMD}}_{k,b}$ in (4) which also relies on the artificially degenerated bootstrap processes, behaves similarly to $V_{b2}$. In the alternative scenario where $\{Y_i\}$ was taken from a distribution with the same covariance structure but with the mean set to $\mu = \begin{bmatrix} 2.5 & 0 \end{bmatrix}$, the Type II error for all tests was zero.

**Pitch-evoking sounds**   Our second experiment is a two sample test on sounds studied in the field of pitch perception [19]. We synthesise the sounds with the fundamental frequency parameter of treble C, subsampled at 10.46kHz. Each $i$-th period of length $\Omega$ contains $d = 20$ audio samples

at times $0 = t_1 < \ldots < t_d < \Omega$ – we treat this whole vector as a single observation $X_i$ or $Y_i$, i.e., we are comparing distributions on $\mathbb{R}^{20}$. Sounds are generated based on the AR process $a_i = \lambda a_{i-1} + \sqrt{1 - \lambda^2}\epsilon_i$, where $a_0, \epsilon_i \sim \mathcal{N}(0, I_d)$, with $X_{i,r} = \sum_j \sum_{s=1}^d a_{j,s} \exp\left(-\frac{(t_r - t_s - (j-i)\Omega)^2}{2\sigma^2}\right)$. Thus, a given pattern – a smoothed version of $a_0$ – slowly varies, and hence the sound deviates from periodicity, but still evokes a pitch. We take $X$ with $\sigma = 0.1\Omega$ and $\lambda = 0.8$, and $Y$ is either an independent copy of $X$ (null scenario), or has $\sigma = 0.05\Omega$ (alternative scenario) (Variation in the smoothness parameter changes the width of the spectral envelope, i.e., the brightness of the sound). $n_x$ is taken to be different from $n_y$. Results in Table 1 demonstrate that the approach using the wild bootstrapped statistic in (4) allows control of the Type I error and reduction of the Type II error with increasing sample size, while the permutation test virtually always rejects the null hypothesis. As in [21] and the MCMC example, the artificial degeneration of the wild bootstrap process causes the Type I error to remain above the design parameter of $0.05$, although it can be observed to drop with increasing sample size.

**Instantaneous independence**  To examine instantaneous independence test performance, we compare it with the Shift-HSIC procedure [8] on the 'Extinct Gaussian' autoregressive process proposed in the [8, Section 4.1]. Using exactly the same setting we compute type I error as a function of the temporal dependence and type II error as a function of extinction rate. Figure 1 shows that all three tests (Shift-HSIC and tests based on $V_{b1}$ and $V_{b2}$) perform similarly.

**Lag-HSIC**  The KCSD [4] is, to our knowledge, the only test procedure to reject the null hypothesis if there exist $t,t'$ such that $Z_t$ and $Z_{t'}$ are dependent. In the experiments, we compare lag-HSIC with KCSD on two kinds of processes: one inspired by econometrics and one from [4].
In lag-HSIC, the number of lags under examination was equal to $\max\{10, \log n\}$, where $n$ is the sample size. We used Gaussian kernels with widths estimated by the median heuristic. The cumulative distribution of the $V$-statistics was approximated by samples from $nV_{b2}$. To model the tail of this distribution, we have fitted the generalized Pareto distribution to the bootstrapped samples ([23] shows that for a large class of underlying distribution functions such an approximation is valid). The first process is a pair of two time series which share a common variance,

$$X_t = \epsilon_{1,t}\sigma_t^2, \quad Y_t = \epsilon_{2,t}\sigma_t^2, \sigma_t^2 = 1 + 0.45(X_{t-1}^2 + Y_{t-1}^2), \quad \epsilon_{i,t} \overset{i.i.d.}{\sim} \mathcal{N}(0,1), \quad i \in \{1,2\}. \quad (6)$$

The above set of equations is an instance of the VEC dynamics [2] used in econometrics to model market volatility. The left panel of the Figure 2 presents the Type II error rate: for KCSD it remains at 90% while for lag-HSIC it gradually drops to zero. The Type I error, which we calculated by sampling two independent copies $(X_t^{(1)}, Y_t^{(1)})$ and $(X_t^{(2)}, Y_t^{(2)})$ of the process and performing the tests on the pair $(X_t^{(1)}, Y_t^{(2)})$, was around 5% for both of the tests.
Our next experiment is a process sampled according to the dynamics proposed by [4],

$$X_t = \cos(\phi_{t,1}), \qquad\qquad \phi_{t,1} = \phi_{t-1,1} + 0.1\epsilon_{1,t} + 2\pi f_1 T_s, \quad \epsilon_{1,t} \overset{i.i.d.}{\sim} \mathcal{N}(0,1), \quad (7)$$

$$Y_t = [2 + C\sin(\phi_{t,1})]\cos(\phi_{t,2}), \quad \phi_{t,2} = \phi_{t-1,2} + 0.1\epsilon_{2,t} + 2\pi f_2 T_s, \quad \epsilon_{2,t} \overset{i.i.d.}{\sim} \mathcal{N}(0,1), \quad (8)$$

with parameters $C = .4$, $f_1 = 4Hz$, $f_2 = 20Hz$, and frequency $\frac{1}{T_s} = 100Hz$. We compared performance of the KCSD algorithm, with parameters set to vales recommended in [4], and the lag-HSIC algorithm. The Type II error of lag-HSIC, presented in the right panel of the Figure 2, is substantially lower than that of KCSD. The Type I error ($C = 0$) is equal or lower than 5% for both procedures. Most oddly, KCSD error seems to converge to zero in steps. This may be due to the method relying on a spectral decomposition of the signals across a fixed set of bands. As the number of samples increases, the quality of the spectrogram will improve, and dependence will become apparent in bands where it was undetectable at shorter signal lengths.

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
