[Reviews · NeurIPS 2014]

Submitted by Assigned_Reviewer_13

The authors propose a bootstrap method for constructing statistical tests on stochastic processes when temporal dependencies exist within the samples. The proposed method is based on a dependent wild bootstrap technique [18]. The main contribution of the paper is to show that this technique produces consistent tests for time series that have an Type I error approaching the desired confidence parameter \alpha and a Type II error approaching zero. The authors use the wild bootstrap technique to construct a two-sample test and an independence test that are evaluated in different experiments. The independence test is applied to testing for independencies at the same time step and across multiple time lags.

Quality:

The paper technically sound, although I did not verified the proofs. The experiments seem to validate the claims made by the authors regarding the strong performance of the tests based on the dependent wild bootstrap technique. One thing that I miss is some description of the computational costs of the proposed approach. It seems that it does not have any additional cost when compared to other possible alternatives such as the permutation test described in [7].

Clarity:

This is the weakest point of the paper. The paper seems to be directed to a reader with a very strong background in statistics. Most of the people in the machine learning community are unlikely to be familiar with concepts such as \tau-mixing, the wild bootstrap method or the dependent wild bootstrap method. The authors should have included an intuitive and clear description of these elements. I had a hard time looking for better descriptions of these elements in the corresponding references, even though I am particularly familiar with concepts such as kernel embedings, MMD, HSIC and V-statistics. My opinion is that the paper seems to be addressed to the statistics community instead of to the machine learning community.

Originality:

It is not clear how original this work is. It seems that the authors are only applying the dependent wild bootstrap method (already described in [18]) to the V-statistics for MMD and HSIC. The authors should state more clearly what are the major novel contributions.

Significance:

The problem addressed by the authors is of high relevance to the machine learning community. There has been a lot of recent work on statistical tests for independence and also on two-sample tests. These are important problems in data modelling and machine learning. The approach proposed by the authors seems to produce consistent tests in many important cases of practical importance (processes with serial dependence) where previous tests were failing.
Summary: The authors present a relevant way to produce consistent tests for processes with serial dependence based on the dependent wild bootstrap technique. The experiments support this. However, the paper is very technical and seems to be addressed to an audience with a very strong background on statistics. It is also not clear how novel the authors' contribution is since they seem to just apply the dependent wild bootstrap method [18] to specific V-statistics for MMD and HSIC.

Submitted by Assigned_Reviewer_16

In this paper, which is in the general context of nonparametric testing based on Hilbert space embeddings, a novel Type 1 error control calibration method for nonparametric hypothesis tests of two weakly dependent samples/processes is proposed. The paper strongly builds on recent work in mathematical statistics [18,25] regarding the wild bootstrap method. The main contribution consists in showing that V statistics (of any order) and in particular those based on RKHS embeddings (including MMD and HSIC) can be asymptotically consistent even for weakly dependent processes (tau mixing). The approach is evaluated in terms of the achieved Type 1 and 2 error rates on syntethic, audio, and Gibbs sampler data.

First of all, the paper is very well written and concise, and despite the high complexity, relatively easy to follow. The paper addresses a the very relevant topic of nonparametric statistical testing based on RKHS embeddings of probability distributions, which has been a vivid topic in ML in recent years. The main contributions of the paper from my point of view are:
- introducing the wild bootstrap method in ML (has solely been used in statistical literature up to now)
- proving that the wild bootstrap yields asymptotic consistency under weak dependence assumptions (tau mixing), for both the null and alternative hypotheses
- Indicating the empirical usefulnesss

The strongest parts of the paper are the methodological contribution and innovative idea, and of course the strong theoretical part, establishing deep theoretical foundations of the suggested testing procedure. The analysis, which is very sounds, is general enough to apply also to V statistics beyond the common ones in ML (i.e., beyond HSIC and MMD). The theory alone would make a good NIPS contribution, but in addition there is also a empirical analysis, which is -- although not entirely convincing when being picky (see detailed comments below) -- promising and shows superior performance of the method in comparison to appropriate baselines.

In summary, this is a novel methodology for a relevant NIPS topic, which is addressed in deep theoretical detail, and included a promising empirical analysis. A clear accept.

Minor, quite picky comments:
- Introduce probability operator \Epsilon or use standard notation like \mathbb P
- Use of the word "control": please correct me if I am wrong, but in my understanding the type 1 error is controlled if the truly measured type 1 error is *smaller or equal* to the desired level. In several experiments (e.g., MCMC Vb1) the actual rate is slighly higher than the desired one, which would strikly speaking mean the error is not entirely controlled (if I get the terminology right).
- Introduction of the wild bootstrap at the end of Sect 2: this is very technical, but nobody in ML knows this quantity. Show an illustration, what does it do intuitively? Example of how to choose the wild bootstrap process: for instance, the statistical learning theory reader might wonder whether it makes sense to use a correlated Rademacher or Gaussian process here.
- Beginning of Sec 4: "This quantity". Let's say I wouldn't know anything about MMD: it is unclear to which term "this quantity" refers.
- Top of page 5: "[18] and 2" -> [18] and [25] ?
- experiments: make clearer what the difference between MMD hat and Vb1 and Vb2 is. Also note that the Type 1 error is not controlled, it is just closer to the desired error
- For a sound empirical analysis, you could use ROC curves. It is neccessary to compare the type 1 error and the power / type 2 error at various level simultaeoulsy for a fair comparision. So:
1. at some places in the text (e.g., end of page 7) you mention the measured type 2 error. This is important and I would suggest mentioning this also in the captions of the figures.
2. What does Figure 1 tell me? The methods could be better evaluated if you plotted a roc or pr curve or something like that, I would think.
3. For the same reason: figure 2 and end of page 8. "was around 5%". This is a little vague and again this should be mentioned in the caption of figure 2.
Same page: "The type... was equal or lower ..." This is again vague. A roc or pr curve would shed light on this.
- The Bonferoni correction for HSIC-Lag will be too conservative under the alternative hypothesis. Is there a way to wrap a (wild) bootstrap around the whole lagged kind of procedure / come up with a joint statistics, which is calibrated via wild bootstrap?
Summary: In summary, this is a novel methodology for a relevant NIPS topic, which is addressed in deep theoretical detail, and included a promising empirical analysis. A clear accept.

Submitted by Assigned_Reviewer_29

This paper shows how the wild bootstrap can be used to construct nonparametric hypothesis tests that are based on kernel distribution embeddings. Previous work has shown how such tests can be constructed when samples are i.i.d. using a permutation test or a closed form approximation of the asymptotic null distribution. In the case of time series, however, the asymptotic distribution is more complicated and the same approximation is not appropriate and permuting samples breaks the temporal independence so neither approach is appropriate for constructing tests in this context. Other previous work proposed an alternative approximation of the null distribution by shifting one signal against another, but this procedure only applies to independence testing with time series (it cannot be used to construct a two sample test to compare the marginal distributions of two time series).

The authors show how the wild bootstrap procedure can be used to simulate from the null distribution of any test statistic that can be cast as a V-statistic, which includes most of the popular kernel distribution embeddings tests, e.g. the Minimum Mean Discrepancy (MMD), a two sample test, and Hilbert Schmidt Independence Criterion (HSIC), an independence test, and that this leads to consistent tests for time series. The authors then use this result to construct a two sample MMD test and an instantaneous and lagged HSIC independence test for time series.

The authors then show how the new MMD test for time series can be used to diagnose the convergence of an MCMC chain and compare the pitches of sounds synthetically generated by AR processes. They also compare the performance of the instantaneous and lagged HSIC to existing methods on synthetic time series data. In each case, the wild bootstrap methods perform comparable to or better than competing methods.

This paper is generally well written and clear. The results are quite rigorous and appear to be sound (though I have not attempted to check all of the details in the proofs). The work is clearly well motivated. The authors provide multiple examples of how this work could be applied in different application areas and as a diagnostic tool for MCMC. It is original. To my knowledge, this provides the first kernel two sample test for time series and may be useful in other kernel tests. It's also original in that it provides an alternative to the permutation test that is frequently used with these types of methods.

There are some areas where the authors could be a bit less terse. I was unfamiliar with the wild bootstrap before reading this paper and had to look it up because the authors description didn't really explain it. Since this is the basis of the paper, a slightly more thorough explanation might be useful since I don't think this flavor of bootstrap is well known. Also for tau-mixing, a very rigorous formal definition is given, but I'm not sure I quite understand what the implications of this definition may be. I'm guessing it is given for completeness and to reference in the proofs, but if there are any unexpected implications that result from using this form of mixing, they should probably be spelled out more explicitly.

I liked the applications and thought the demonstrated the performance of the new methods well. However, one example with non-synthetic data would be a nice addition.
Summary: This paper shows how the wild bootstrap can be used to sample from the null distribution of any test statistic that can be cast as a V-statistic, which includes the popular kernel distribution embedding based MMD and HSIC tests, and that the resulting test is consistent. The paper is well written, technically sound, well motivated by examples of applications to time series testing and diagnosing the convergence of MCMC, and original in that it provides the first kernel two sample test and lagged kernel independence test for time series.
Author Feedback
Author rebuttal: We thank the reviewers for their comments, and for their kind assessment of our work. We agree with reviewers 1-3 regarding their suggested improvements to the clarity of the presentation, which we will implement in a final version.

Multiple reviewers asked for an introduction and illustration of tau mixing and the wild bootstrap. We are happy to add a brief overview of the relevant ideas as a section in the appendix.

Specific comments, Rev. 1

As a comment on computational cost: as in the original HSIC and MMD tests for i.i.d. data, the computational cost of the wild bootstrap approach scales quadratically in the number of samples, and linearly in the number of bootstrap iterations (in the i.i.d. case, these were permutations of the data). The main alternative approaches are the lagged bootstrap of [7], which has the same scaling with data and number of bootstraps, and the spectrogram approach of [4] (note, however, that both these alternative approaches apply only to the independence testing case). The cost of [4] is comparable to our approach, however the statistical power of [4] was much weaker on the data we examined.

In terms of the original technical contributions of the paper: The results in [18] apply to statistics having two arguments. Theorem 1 extends these results to the case of m arguments, and proposition 1 further extends these results to multi-sample statistics. Further, Theorem 2 shows consistency of the statistic under the alternative, which was shown only for particular examples in [18], and not as a general result. We will clarify these points in the final version.

Specific comments, Rev. 2

"type I control": we agree with the reviewer regarding our use of this term. Indeed, our analysis applies to the type I error in the asymptotic regime.

Correlated Rademacher processes have been used in the wild bootstrap literature, however it would need to be checked that these processes satisfy the requirements for the proof of Leucht and Neumann, or alternatively, that this proof could be adapted for the correlated Rademacher case. We use a particular Gaussian process (the O-U process) as recommended in [18], which does satisfy the relevant mixing conditions.

Improved clarity in experimental results: we will improve the presentation in our final manuscript.

Replacement of Bonferoni correction: we have been looking into a couple of ideas, including the asymptotics of the maximum of the statistics across lags and the sum. However the analysis remains difficult, and is ongoing. The idea of using a wild bootstrap on top of these statistics is also an interesting one, and we will look into it. That said, the Bonferroni correction, despite being conservative, still performs well in the examples we looked at.